# Micro-Nanometer Particle Composition and Functional Design of Surface Nano-Structured Ammonium Polyphosphate and Its Application in Intumescent Flame-Retardant Polypropylene

**DOI:** 10.3390/nano12040606

**Published:** 2022-02-11

**Authors:** Xiaolu Wu, Zhaolu Qin, Xiang Zhang, Zhenfei Yu, Wenchao Zhang, Rongjie Yang, Dinghua Li

**Affiliations:** 1National Engineering Research Center of Flame Retardant Materials, School of Materials, Beijing Institute of Technology, Beijing 100081, China; 3120185580@bit.edu.cn (X.W.); zwc18@bit.edu.cn (W.Z.); yrj@bit.edu.cn (R.Y.); dli@bit.edu.cn (D.L.); 2School of Mechanical Engineering, Beijing Institute of Technology, Beijing 100081, China; 6120170012@bit.edu.cn; 3School of Materials, Beijing Institute of Technology, Beijing 100081, China; yuzhenfei@bit.edu.cn

**Keywords:** ammonium polyphosphate, aluminium isopropoxide, polypropylene, intumescent flame retardant, micro-nano composite particles

## Abstract

A novel composite and functional micro-nanometer particle is designed by the hydrolysis of aluminium isopropoxide on the surface of ammonium polyphosphate (APP) to prepare surface nanostructured ammonium polyphosphate (NSAPP). NSAPP is characterised by XPS, XRF, SEM, water solubility tests, and TGA. Results indicate that nanosized aluminium hydroxide is deposited on the surface of NSAPP, which enhanced its water resistance and thermostability. Then, APP and NSAPP coupled with dipentaerythritol (DPER) is used for the flame retardant of polypropylene (PP). The limiting oxygen index (LOI) value of the PP/DPER/NSAPP composite is higher than that of PP/DPER/APP. Besides, the UL 94 vertical burning test of PP/DPER/NSAPP composites can reach the V-0 rating easily. According to the study of the combustion behaviour of FR-PP composites, NSAPP contributes to form a dense and multi-layered char in the combustion process. Thus, such an intumescent char with a ceramic-like, continuous, and dense structure over the PP matrix protects the underlying matrix and enhances the thermal stability of the condensed phase, thereby improving the flame retardant performance of FR-PP.

## 1. Introduction

As one kind of general plastic, polypropylene (PP) is widely used in each domain of industrial production, such as in automobiles, electronics, furniture, architectural materials, and so on [1,2,3]. Due to the inflammable nature of PP, the melting drip is often accompanied by burning PP, which limits its application. This problem can be solved by using flame retardant additives into a PP matrix [4,5,6,7,8]. There have been demonstrated successes by adding halogenated flame retardants into a PP matrix. However, with the increase of consciousness for environment protection, the application of halogenated flame-retardants has been limited. So, as an alternative, halogen-free alternatives have good potential for practical applications [9].

In the field of halogen-free flame-retardants, the P-N intumescent flame retardant (IFR) has attracted much attention in recent years due to its halogen free, low smoke, and low toxicity in burning [10,11,12]. IFR has been known for many years and is recognized as one of the effective ways to realise halogen-free flame retardants. Generally, a traditional IFR consists of three components: acid, carbon, and a gas source [13,14]. Ammonium polyphosphate (APP) is an important inorganic flame retardant, which is used as one of the main ingredients of IFR. Unfortunately, APP based IFR systems are unsustainable due to some disadvantages of APP, for example, the poor water resistance and organics miscibility [15]. The migration and exudation of IFR may occur under humid environments, which can lead to the decrease in flame retardancy. In addition, the smooth and flat surface of APP appears to be against the stable interface connection between the polymer matrix and particle surface [16]. Multitudinous additives and pathways have been tried in the surface modification of APP to overcome the above disadvantages [17,18]. Traditional strategies for surface modification of APP are by coupling agents [19], microencapsulation technologies [20,21], or surfactants [22]. Among various modification technologies, the surface modification of APP with another flame retardant directly has not been reported as yet.

Over the recent years, with the rapid development of nanotechnology, the addition of nano-filler into a polymer matrix has become increasingly mature and efficient, as an important means to improve the flame retardant performance of materials [23,24]. As a common inorganic filler in macromolecular material, nano aluminium hydroxide (ATH) shows the effect of flame retardation and smoke inhibitory simultaneously [25]. However, the dispersibility of pure nanoparticles in a polymer matrix is poor and they are easy to agglomerate. In addition, nano flame retardant system shows excellent performance in the cone calorimeter test, but it is relatively inefficient in traditional combustion tests, such as LOI and UL-94 [26]. Therefore, nanosized ATH powders have not been very effective in polymers as a result of the poor dispersion [27].

The synergistic effect between different flame retardants provides a new idea for the development of flame retardant technology. In our previous work, the synergistic effects between APP and nanosized ATH have been reported [28]. The mechanical and flame retardancy of PP/DPER/APP composites were improved by a proper addition of nanosized ATH. Although the positive interaction between APP and nanosized ATH has been proved by the physical blending of these two materials, the inherent disadvantages, such as weak water resistance of APP, still limit their application in humid environments. Therefore, based on these problems, a novel functional micro-nanometre particle (NSAPP) was prepared by the hydrolysis of aluminium isopropoxide (AIP) on the surface of APP. XPS analysis shows that ATH was coated on the surface of the APP particles. SEM results show that the surface roughness of NSAPP increases significantly. The effects of NSAPP on compatibility, thermostability, and the flame-retardant properties of the FR-PP composites are also investigated.

## 2. Experimental

### 2.1. Materials

Polypropylene (PP, T30S) was provided by the China Petroleum & Chemical Corporation (Beijing, China). Ammonium polyphosphate (APP) was provided by the National Research Center of Flame Retardant Materials Engineering (Beijing, China). Aluminium isopropoxide (AIP) and aluminium hydroxide (ATH) were purchased from the Beijing Tongguang Fine Chemicals Company (Beijing, China). Dipentaerythritol (DPER) was provided by Shandong Pulisi Chemical Co., Ltd. (Zibo, China). Anhydrous ethanol, isopropanol, and acetic acid were purchased from Meryer (Shanghai, China) Chemical Technology Co., Ltd. The antioxidant 1010 and antioxidant 168 were supplied by Switzerland Ciba Chemistry (China) Co., Ltd. (Shanghai, China).

### 2.2. Preparation of NSAPP Particles

First, 50 g of APP was dispersed in 300 mL anhydrous ethanol. Then a certain amount of AIP was dissolved in 100 mL of isopropanol. After that, the AIP solution was added into the above reaction system. The mixed solution was stirred and kept at 80 °C for 30 min. Then, 50 mL of distilled water was added into the mixture dropwise. The reaction system was kept at 80 °C for 8 h. Finally, the final product was filtered, washed with anhydrous ethanol, and dried at 100 °C. According to the dosage of AIP (2 g, 4 g, 6 g, and 8 g), samples were named as NASPP-1, NSAPP-2, NSAPP-3, and NSAPP-4, respectively.

### 2.3. Preparation of Flame Retardant PP Composites

The flame retardant PP composites were processed by a twin-screw extruder (SHJ-20) (Nanjing Giant Machinery Co., Ltd., Nanjing, China). The processing temperature of each section was 175 °C, 180 °C, 185 °C, 185 °C, 180 °C, and 170 °C. The feed rate was 11 rpm and the screw speed was 20 rpm. Test samples were prepared by an injection moulding machine (HTF80X1). The temperature of each section for the injection moulding machine was 195 °C, 195 °C, 190 °C, and 185 °C. The formulation of the flame-retardant PP composites is given in Table 1.

### 2.4. Experimental Analysis

Water solubility was measured as follows: add 10 g of APP or NSAPP into distilled water (100 mL). The mixture was stirred for 1 h at different temperatures (25 °C, 50 °C, and 75 °C) and centrifuged for 20 min. Then, 20 mL of supernatant liquid was removed and dried to constant weight. Then, the water solubility of APP and NSAPP can be calculated.

The morphological characterisation of APP, NSAPP, char residue, and the fractured section of FR-PP composites was carried out using a Scanning Electron Microscope (Hitachi, S4800) (Hitachi, Japan).

The X-ray photoelectron spectroscopy (XPS) data were obtained by a Perkin Elmer PHI 5300 ESCA system (Waltham, MA, USA) at 250 W (12.5 kV at 20 mA) under a vacuum better than 10^−6^ Pa (10^−8^ Torr). C_1s_ spectra was shifted to standard positions of 284.5 eV.

Thermogravimetric analysis was measured on a thermal analyser (NETZSCH 209 F1. Selb, Germany) from 40 °C to 800 °C under N_2_ atmosphere. The heating rate was 10 °C/min with a gas flow rate of 50 mL/min. The mass of each sample was about 10 mg.

Cone calorimeter tests were performed according to the ISO 5660 protocol at an incident radiant flux of 50 kW/m^2^ with square specimens of 100 mm × 100 mm × 3.2 mm.

LOI values were obtained with an FTA-II instrument (Rheometric Scientific Ltd. (Dorset, UK) and specimen dimensions of 118 mm × 6.5 mm × 3 mm, according to the ASTM D 2863 standard. The vertical test was carried out on a CZF-5A-type instrument (Jiangning Analysis Instrument Company, Nanjing, China) according to the UL 94 test standard. The specimens used were of dimensions 125 mm × 13 mm × 3.2 mm.

The rheological behaviour of FR-PP composites was measured by a strain rheometer (RS300, Thermo HAAKE Scientific, Waltham, MA, USA). Measurements were performed in the plate-plate configuration with a gap of 1 mm. The sensor type was PP20 Ti. An oscillation frequency sweep was completed on the PP at 200 °C. The range of ω was 0.01–100 rad/s1, τ = 1 Pa.

## 3. Results and Discussion

### 3.1. XPS Analysis of APP and NSAPP

The components of APP and NSAPP were discussed based on the results of XPS analysis. The XPS spectra of APP and NSAPP were shown in Figure 1. According to the molecular formula of APP, the binding energy of O_1s_ (530.6 eV), N_1s_ (400.8 eV), P_2p_ (134.5 eV), and P_2s_ (190.9 eV) were detected. While for NSAPP, the binding energy of Al_2p_ (74.2 eV) and Al_2s_ (119.1 eV) were observed, which suggests the presence of Al on the surface of the NSAPP because of the existence of aluminium hydroxide [29].

Table 2 presents the surface elemental composition of NSAPP and APP particles. The N, O, and P elemental contents of APP are 15.49, 43.83, and 16.22 wt.%, respectively. For NSAPP particles, the N and P elemental contents decreased, while the Al and O elemental contents increased with the gradually increasing dosage of AIP.

X-ray fluorescence spectroscopy (XRF) has been widely applied to the qualitative and quantitative analysis of different elements and materials [30,31]. Here, a quantitative analysis method of NSAPP particles was established by XRF to verify the specific content of the surface nanosized ATH. A series of standard samples was prepared by blending neat APP with commercial ATH in different mass ratios. The weight percentage of ATH ranged from 0% to 5% and the content variation interval was 0.5%. The calibration curve of ATH content (X) and Al measured intensity (Y) was established based on the analysis of these standard samples using XRF, as shown in Figure 2. After testing and analysing the NSAPP particles, the measured contents of surface nanosized ATH for NSAPP-1, NSAPP-2, NSAPP-3, and NSAPP-4 particles are 1.13%, 2.17%, 2.96%, and 4.08%, respectively. According to the dosage of AIP, the theoretical contents of surface nanosized ATH for NSAPP-1, NSAPP-2, NSAPP-3, and NSAPP-4 particles are 1.5%, 2.29%, 4.38%, and 5.76%, respectively. Predictably, the experimental data are slightly lower than theoretical value.

### 3.2. SEM of APP and NSAPP Particles

The microstructure of APP and NSAPP is observed with SEM in Figure 3. It can be seen that the particles of APP are club shaped with a smooth and regular surface. However, NSAPP particles possess a rough surface due to plentiful nanoparticles deposited on the surface of NSAPP. Besides, the size of these nanoparticles is less than 100 nm, which can be demonstrated from the SEM photomicrographs of Figure 4. In addition, the water resistance of APP and NSAPP has been tested using water contact angle (WCA) measurements. Result shows that NSAPP particles are more hydrophobic than that of APP. The WCA of APP is 7.8°, whereas the WCA of NSAPP particles is higher than the neat APP, suggesting that the surfaces of the NSAPP particles change from hydrophilic to hydrophobic. This might be down to the protective effect of nanosized ATH, which deposited on the surface of NSAPP. This result is found in good coincidence with the water solubility results.

### 3.3. Water Solubility of APP and NSAPP Particles

The solubility of APP and NSAPP at varied temperatures is shown in Figure 5. As we can see, the solubility of APP is 0.6895 g/100 mL at 25 °C, 1.5645 g/100 mL at 50 °C, and 3.8390 g/100 mL at 75 °C, respectively. Apparently, the solubility of APP increases with increasing temperature. For NSAPP particles, especially NSAPP-3, the water solubility is 0.2665 g/100 mL at 25 °C, 0.4135 g/100 mL at 50 °C, and 0.7325 g/100 mL at 75 °C, respectively. The result shows NSAPP presents excellent water resistance with that of APP. With the increasing concentration of AIP, water solubility of NSAPP is obviously declined. After the concentration of AIP reaches a certain level, the water solubility of NSAPP will no longer decreases any more.

### 3.4. Thermogravimetric Analysis

Figure 6 presents the thermogravimetric (TG) and differential thermogravimetric (DTG) curves of APP and NSAPP particles in the atmosphere of nitrogen. Table 3 shows details about the data. T_5%_, R*_max_*, and T*_max_* are defined as the initial degradation temperature, the maximum weight loss rate, and the temperature at maximum weight loss rate, respectively.

The degradation process of APP has already been proved by many researchers [32,33]. From Figure 6, it can be seen that the thermal degradation process of APP can be mainly divided into two stages. The first stage occurred between 250–400 °C, corresponding to the release of ammonia and water. The resulting degradation products crosslink in highly condensed polyphosphoric acid [34]. As shown in Table 3, the T_5%_ for neat APP is 303 °C, while, in the case of NSAPP particles, the T_5%_ is lower than that of neat APP. Moreover, a slight weight loss is detected at about 200 °C due to the dehydration of the surface nanosized ATH on NSAPP particles. The second stage occurred between 500–700 °C, a rapid mass loss was detected, which occurs as a result of the breakdown of the ultraphosphate structure and the formation of P_2_O_5_ [34]. The R*_max_* and T*_max_* of APP are −14.05%/min and 624 °C, respectively. The R*_max_* and T*_max_* of NSAPP are much lower than that of APP. Based on the thermal degradation mechanisms of APP and ATH, it can be inferred that when the temperature is higher than 500 °C, the polyphosphoric acid from the degradation of APP reacted with the aluminium oxide from the degradation of ATH and generated aluminium phosphate-like compounds, which have higher thermal stability. This product reduced the mass loss rate and increased the thermal stability of the residues at high temperatures [28]. Therefore, gas products are allowed to leave the solid phase at high temperature. Moreover, the P-Al-O compound leads to more residue, corresponding to the higher residue weight of NSAPP-4 at 800 °C (29.87%) as compared with that of APP (14.48%).

### 3.5. Morphology, Mechanical Properties, and Rheological Behaviour of FR-PP Composites

Figure 7 shows the morphology of the fracture surface of FR-PP composites. It is evident from Figure 7 that NSAPP particles are closely combined with the PP matrix. On the contrary, it’s easy to see the grooves and gaps which exist on the fracture surface of PP/DPER/APP composites (Figure 8a). The binding status and intensity of the NSAPP and the PP matrix is better than that of neat APP. This is because the nanoparticles coated on the surface of NSAPP create rough surfaces with large surface areas (Figure 8) and thereby increase the contact area between NSAPP and the PP matrix.

As shown in Figure 9, compared with pure APP, the FR-PP composites containing NSAPP particles exhibit increased tensile strength and elongation at break. The tensile strength, in particular, increases from 24.38 MPa (PP/DPER/APP) to 25.55 MPa (PP/DPER/NSAPP-4), and the elongation at break increases from 15.72% to 30.22%. In addition, with the increasing content of ATH on surface of NSAPP particles, both the tensile strength and elongation at break increase significantly, but the rates of increase for the test parameters are slowed. Based on these results, it can be concluded that the right amount of ATH on surface of NSAPP particles significantly improves the mechanical properties of FR-PP. However, the continued increase in the content of ATH does not cause further improvement in the mechanical properties.

Some significant results of the close relationship between the viscoelastic characteristics and flammability properties of thermoplastic polymers were reported [35,36]. Figure 10 shows the viscosity of FR-PP composites versus the shearing rate. Clearly, the viscosity of the PP/DPER/APP composite is lower than that of the PP/DPER/NSAPP composites at different shear rates. It is ascribed that the smooth and regular surface of APP is unhelpful for the hindrance of the entanglement of matrix resin molecules. With the increase of the surface roughness of NSAPP particles, the free volume of PP melt was reduced by the strong interaction between NSAPP particles and the PP molecular chain. The steric resistance of the molecular chain and the apparent viscosity of the polymer melt are increased. This phenomenon is also found and explained in some literatures concerning filled composite materials [37]. PP/DPER/APP composite exhibits a quasi-Newtonian regime during the test, and the shearing stress has little effect on its viscosity, while the PP/DPER/NSAPP composites show a typical shear-thinning behaviour of polymer melts and the values of viscosity increased with the increasing content of the surface nanosized ATH. A higher viscosity is beneficial to form a steadier char under the heat flux during the cone calorimeter test.

### 3.6. LOI and UL 94 Testing of FR-PP Composites

The LOI and UL 94 tests are usually used to evaluate the flame retardancy of polymer materials. As shown in Table 4, the LOI value of PP/DPER/APP is 26.8% and it can only reach the V-1 level in the UL 94 test. For PP/DPER/NSAPP composites, the result of the UL 94 test raised from the V-1 to V-0 level. The LOI values of PP/DPER/NSAPP composites also reflect an obvious improvement. In addition, the self-extinguishing times for PP/DPER/NSAPP composites are obviously reduced with that of the PP/DPER/APP composite. The result reveals the PP/DPER/NSAPP composites show better flame retardancy and char-forming ability. For PP/DPER/NSAPP-3 and PP/DPER/NSAPP-4 composites, with the increased content of ATH on surface of NASPP, a slight drop in flame retardancy for FR-PP composites was observed due to the excessive consumption of APP, which is caused by the reaction between APP and ATH [28,33].

### 3.7. Cone Calorimeter Testing of FR-PP Composites

Results of cone calorimeter tests performed on the PP control and FR-PP composites are summarised in Figure 11. The corresponding combustion parameters, which include the peak of heat release rate (p-HRR), total heat release (THR), time to ignition (TTI), average CO release (avCO), average CO_2_ release (avCO_2_), total smoke production (TSP), and the char residues after the cone test, are listed in Table 5.

The incorporation of APP/DPER in PP shows good flame retardancy at the loading of 25 wt.% and reduces by 40% with the decrease of p-HRR [26]. For an intumescent flame retardant system, the intumescent char may act as a barrier, keeping the underlying matrix from further degradation. For the HRR curve of PP/DPER/APP, a sharp peak occurred at about 260 s, meaning that the protective barrier has apparently failed at that moment. For PP/DPER/NSAPP composites, the HRR curve is flatter indicating that a more effective shielding layer is formed and can prevent the rapid release of heat during combustion and reduce the possibility of a fire. In addition, photographs of the char residues also show that the insulation and heat resistance of the shielding layer is further heightened by adding NSAPP instead of APP. During combustion, the char produced by the PP/DPER/NSAPP-3 composites could prevent the heat transfer, the diffusion of decomposition products, and the release rate of volatile products more effectively, as well as acting as a physical barrier effect. All these lead to the decrease of p-HRR and THR.

## 4. Conclusions

In this study, surface nanostructured ammonium polyphosphate (NSAPP) was prepared by the hydrolysis of aluminium isopropoxide, which proceeded on the surface of APP. Aluminium hydroxide is detected on the surface of NSAPP by the results of XPS and XRF. NSAPP shows better water resistance, thermal stability, and compatibility with the PP matrix than that of APP. Moreover, the addition of NSAPP could improve the flame retardancy of PP than that of APP. When NSAPP/DPER was added to 25 wt.%, the LOI value reached 30.9% and the UL-94 rating raised to V-0 (3.2 mm). Cone calorimeter tests indicate that the pHRR decreased significantly and a better protecting char was formed.

## Figures and Tables

**Figure 1 nanomaterials-12-00606-f001:**
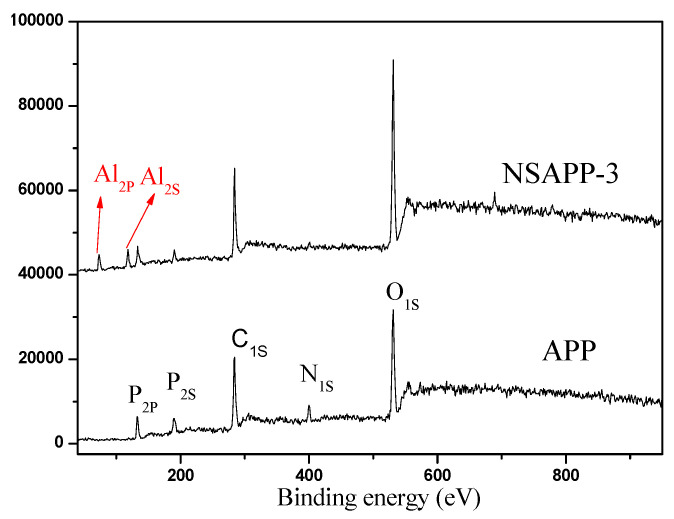
XPS spectra of APP and NSAPP.

**Figure 2 nanomaterials-12-00606-f002:**
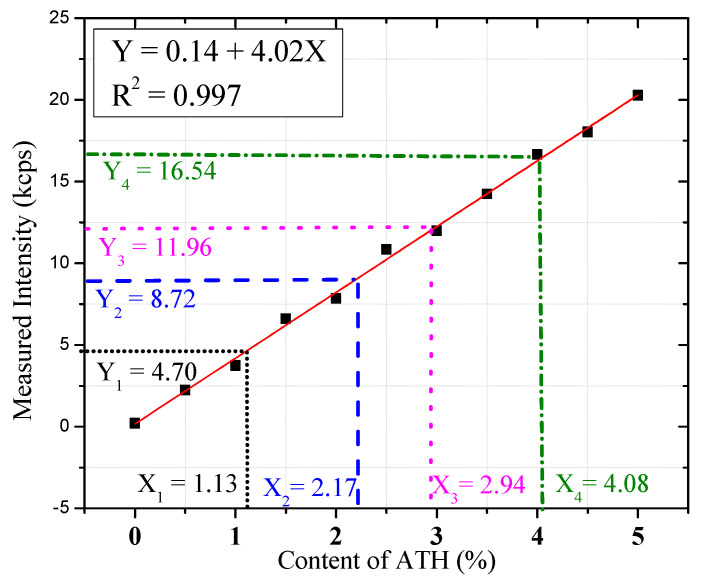
Calibration curves of content of ATH and measured intensity.

**Figure 3 nanomaterials-12-00606-f003:**
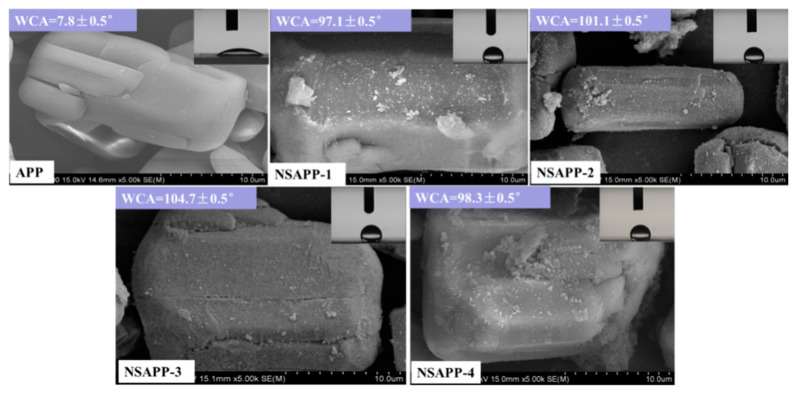
SEM micrographs of APP and NSAPP (×5000).

**Figure 4 nanomaterials-12-00606-f004:**
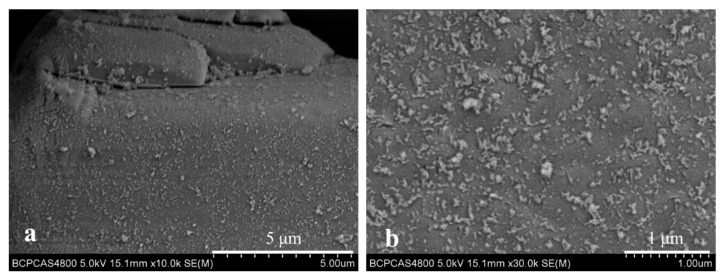
Surface morphology of NSAPP-3: (**a**) ×10,000 and (**b**) ×30,000.

**Figure 5 nanomaterials-12-00606-f005:**
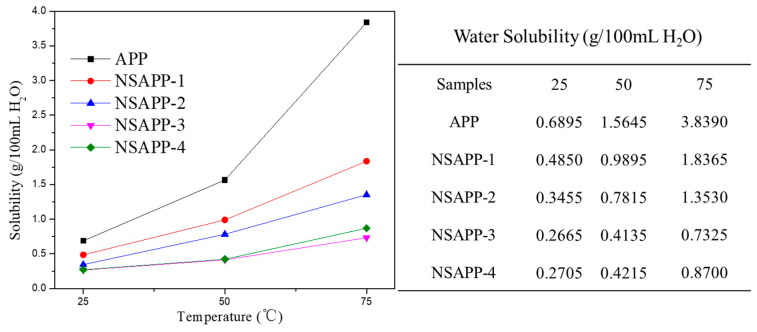
Water solubility of APP and NSAPP particles at different temperatures.

**Figure 6 nanomaterials-12-00606-f006:**
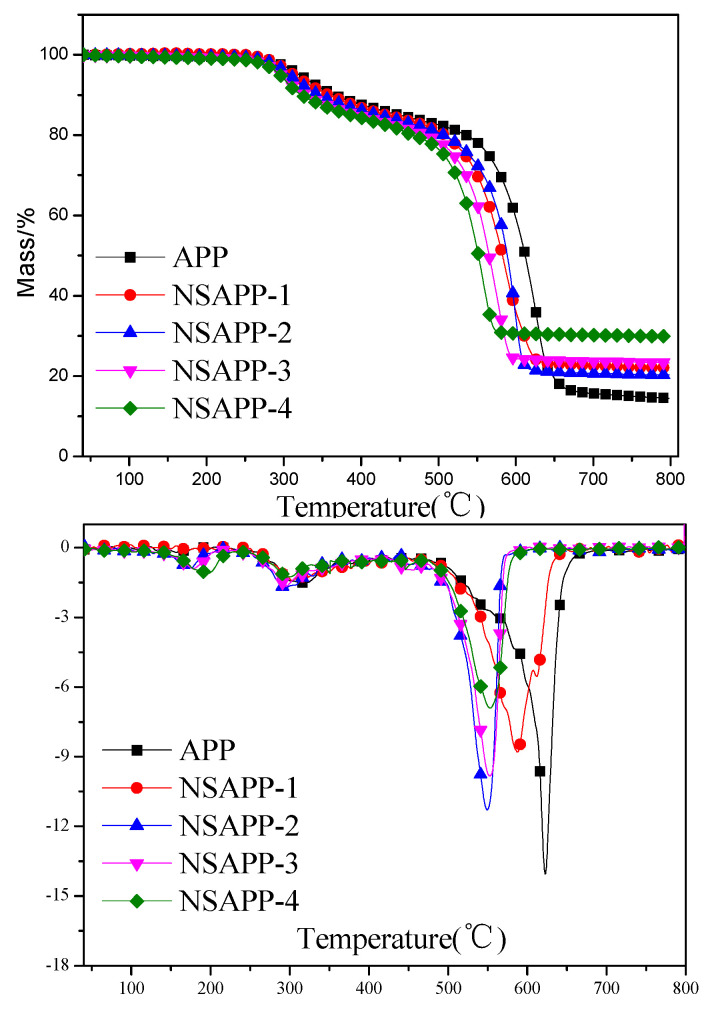
TG and DTG curves of APP and NSAPP particles (N_2_ atmosphere).

**Figure 7 nanomaterials-12-00606-f007:**
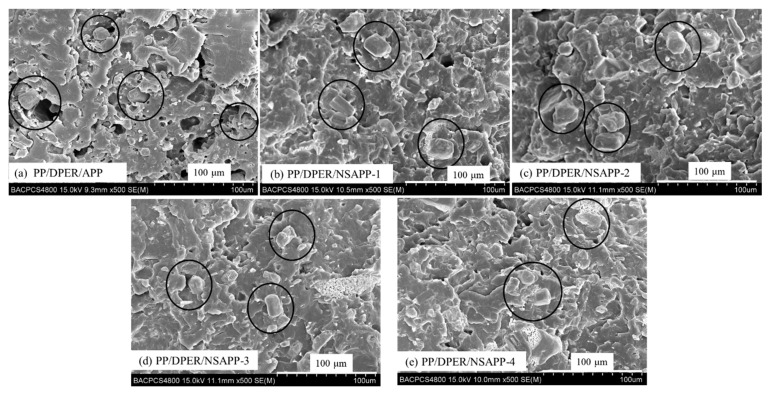
Scanning electron photographs of FR-PP composites fracture surface. (**a**) PP/DPER/APP; (**b**) PP/DPER/NSAPP-1; (**c**) PP/DPER/NSAPP-2; (**d**) PP/DPER/ASAPP-3; (**e**) PP/DPER/ASAPP-4.

**Figure 8 nanomaterials-12-00606-f008:**
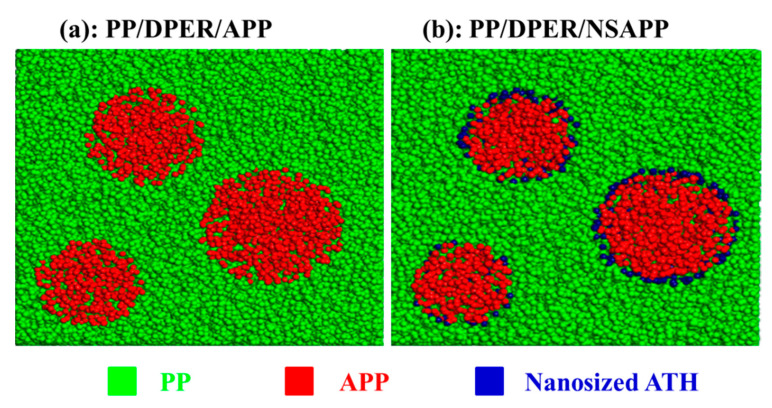
Schematic diagrams of the fracture surface of PP/DPER/APP and PP/DPER/NSAPP composites. (**a**) PP/DPER/APP; (**b**) PP/DPER/NSAPP.

**Figure 9 nanomaterials-12-00606-f009:**
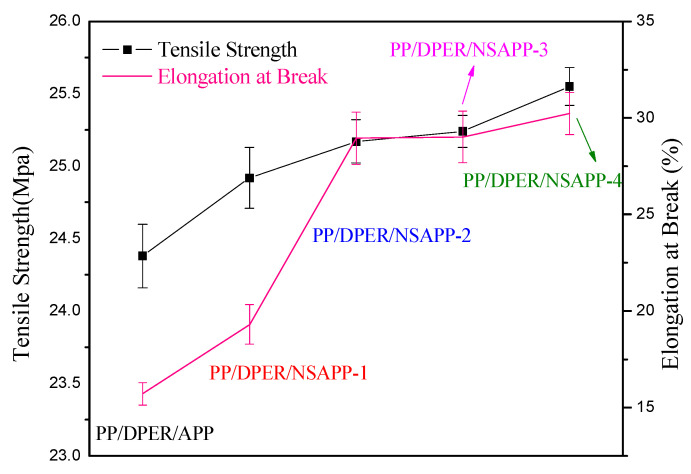
Tensile strength and elongation at break of FR-PP composites.

**Figure 10 nanomaterials-12-00606-f010:**
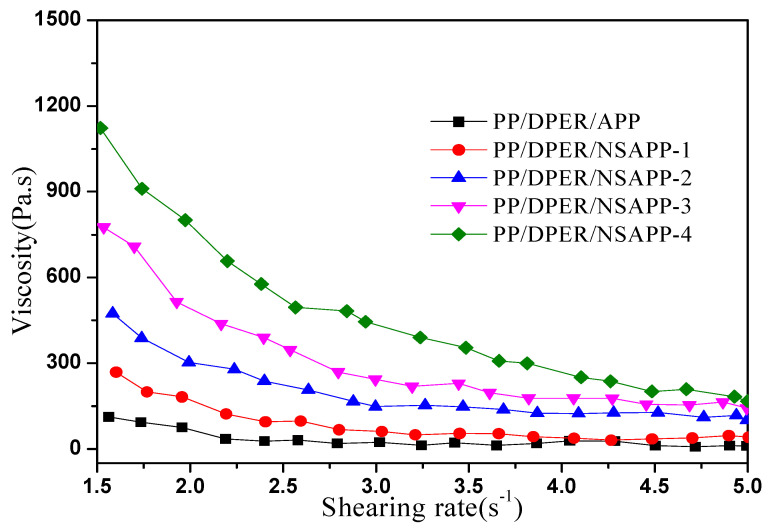
Viscosity of FR-PP versus shearing rate.

**Figure 11 nanomaterials-12-00606-f011:**
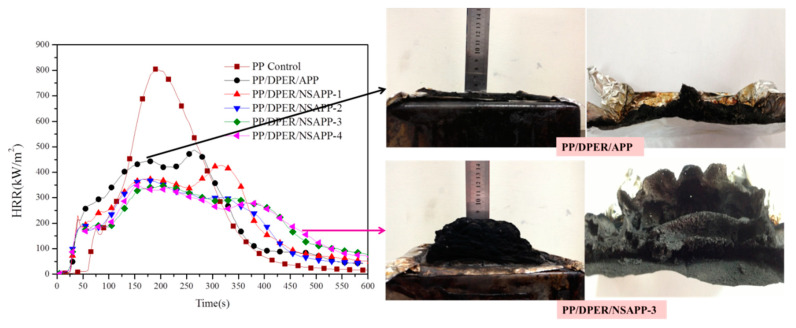
Heat release rate curve and images of the residues of FR-PP composites after the cone calorimetry tests.

**Table 1 nanomaterials-12-00606-t001:** Formulations of flame retardant PP composites (wt%).

Sample	PP	APP	NSAPP-1	NSAPP-2	NSAPP-3	NSAPP-4	DPER	1010	168
PP/DPER/APP	74.7	17.2	-	-	-	-	7.8	0.1	0.2
PP/DPER/NSAPP-1	74.7	-	17.2	-	-	-	7.8	0.1	0.2
PP/DPER/NSAPP-2	74.7	-	-	17.2	-	-	7.8	0.1	0.2
PP/DPER/NSAPP-3	74.7	-	-	-	17.2	-	7.8	0.1	0.2
PP/DPER/NSAPP-4	74.7	-	-	-	-	17.2	7.8	0.1	0.2

**Table 2 nanomaterials-12-00606-t002:** Surface elemental composition of APP and NSAPP.

Samples	Elements Concentration (%)				
	C	N	O	P	Al
APP	24.46	15.49	43.83	16.22	0.00
NSAPP-1	25.74	8.83	46.48	13.35	5.60
NSAPP-2	23.49	6.78	48.53	10.26	10.94
NSAPP-3	22.52	5.31	51.28	7.23	13.66
NSAPP-4	23.38	4.01	53.98	5.11	13.52

**Table 3 nanomaterials-12-00606-t003:** Thermal stability parameters of APP and NSAPP particles.

Sample	T_5%_ (°C)	R*_max_* (%/min)	T*_max_* (°C)	Residue (%)
APP	303	−14.05	624	14.48
NSAPP-1	301	−8.79	587	22.08
NSAPP-2	297	−11.29	546	20.25
NSAPP-3	295	−9.83	551	23.31
NSAPP-4	289	−6.90	553	29.87

**Table 4 nanomaterials-12-00606-t004:** LOI value and UL-94 test results of FR-PP composites.

Samples	LOI (%)	UL-94 (3.2 mm)	t_1_ (s)	t_2_ (s)	Mass Loss (%)
PP/DPER/APP	26.6	V-1	0.8	8.7	8.34
PP/DPER/NSAPP-1	28.7	V-1	0.9	7.2	4.97
PP/DPER/NSAPP-2	30.9	V-0	0.7	2.8	0.98
PP/DPER/NSAPP-3	29.6	V-0	0.7	3.6	1.41
PP/DPER/NSAPP-4	28.5	V-0	0.8	4.3	1.33

**Table 5 nanomaterials-12-00606-t005:** Cone Calorimeter Data of FR-PP composites.

Sample	TTI (s)	p-HRR (kW/m^2^)	avCO (kg/kg)	avCO_2_ (kg/kg)	THR (MJ/m^2^)	TSR m^2^/s
PP Control	35	809	0.05	3.56	144	1297
PP/DPER/APP	23	487	0.10	3.03	136	2036
PP/DPER/NSAPP-1	21	426	0.08	2.93	132	1899
PP/DPER/NSAPP-2	20	368	0.08	2.97	120	1770
PP/DPER/NSAPP-3	21	346	0.08	2.91	123	1914
PP/DPER/NSAPP-4	21	345	0.09	2.95	128	1927

## Data Availability

The data presented in this study are available on request from the corresponding author. The data are not publicly available due to privacy concerns.

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
