# Peer review of "Micro-Nanometer Particle Composition and Functional Design of Surface Nano-Structured Ammonium Polyphosphate and Its Application in Intumescent Flame-Retardant Polypropylene"

_nanomaterials, 2022, doi:10.3390/nano12040606_

Round 1

Reviewer 1 Report

The paper is written more like a technical, not scientific, paper. (style must be improved.)
Besides the environmental issue problem, I would like that authors,  first highlight the study's scientific novelty and what is new from previously published results regarding Al nanoparticles.
Bellow table 1 in Figure caption all used short notation should be described; it's not easy for readers to follow the text with so many brief notations.
In the experimental part, authors should first present structural characterization that confirms that authors make particles, and later stability study  of the particles (regarding water, heat, etc.)
The quality of Figure 5 must be improved, and the size bar must be visible
In the whole text, the discussion part must be enlarged, and results should be discussed (structural, characterization stability, TG and DTG)more in-depth from a physical perspective, and the relationship between structure and physical properties must be established and highlighted.
In the end, I recommend the major revision of this draft.

Author Response

1.The paper is written more like a technical, not scientific, paper. (style must be improved.) Besides the environmental issue problem, I would like that authors, first highlight the study's scientific novelty and what is new from previously published results regarding Al nanoparticles.

Reply: Thank you very much for your advice. According to your advice, we have carefully checked the other related works. The introduction of this manuscript was extended and the differences with other works were mentioned. Besides, the content of this manuscript was also adjusted and more discussions about TG, DTG and mechanical properties were also added to increase scientificity and reliability of this manuscript.
2. Bellow table 1 in Figure caption all used short notation should be described; it's not easy for readers to follow the text with so many brief notations.

Reply: Thank you very much for your advice and comment. We have carefully checked the manuscript. Now, all acronyms (such as ATH, APP, DPER and PP) in Table 1 were defined in full before being used in the revised manuscript. Seen in Section “2.1. Materials
3. In the experimental part, authors should first present structural characterization that confirms that authors make particles, and later stability study of the particles (regarding water, heat, etc.)

Reply: Thank you very much for your advice and comment. According to your advice, we have carefully checked the manuscript and the content of the article has been adjusted.
4. The quality of Figure 5 must be improved, and the size bar must be visible

Reply: Thank you for your comments, we have supplement the new SEM microphotographs of NSAPP and the size bar was also noted in the figure, see in “Figure 4”.

  1. In the whole text, the discussion part must be enlarged, and results should be discussed (structural, characterization stability, TG and DTG) more in-depth from a physical perspective, and the relationship between structure and physical properties must be established and highlighted.
    Reply: Thank you very much for your advice. According to your advice, the discussions part of thermogravimetric analyses was added in the revised manuscript. Besides, the mechanical properties of FR-PP composites were also added in the revised manuscript. Indeed, the result of the mechanical property is very helpful.

Reviewer 2 Report

Paper is very interesting and has scientific soundness. Minor requests are listed below:

Figure 9: No explanation is given for the behaviour of PP/DPER/APP and obtained low results.

Table 4: In the disscussion of obtained results the influence of AIP concentration should be commented. What is the explanation for lower results of PP/DPER/NSAPP-4 samples?

Author Response

Paper is very interesting and has scientific soundness. Minor requests are listed below:

  1. Figure 9: No explanation is given for the behaviour of PP/DPER/APP and obtained low results.

Reply: Thank you very much for your advice and comment. The explanation for the behaviour of PP/DPER/APP and obtained low results was added in the revised manuscript. Related references are also cited to support the explanations.

  1. Table 4: In the disscussion of obtained results the influence of AIP concentration should be commented. What is the explanation for lower results of PP/DPER/NSAPP-4 samples?

Reply:  Thank you very much for your advice and comment. The explanation for lower results of PP/DER/NASPP-4 was added in the revised manuscript. Related references are also cited to support the explanations.

Round 2

Reviewer 1 Report

I still have doubts regarding this paper regarding the importance and novelty. However, the authors replied correctly to all referee's comments and based on that fact; I am recommending the acceptance of the revised draft in unchanged form.